# Maintain electrical power system model quality with Shape Constraints

Merlin Bögershausen[1,3], Oliver Scheufeld[1], Christoph
Lange[2,3][0000−0001−9879−3827], and Oya Deniz Beyan[2,3][0000−0001−7611−3501]

[1] SOPTIM AG<first>.<last>@soptim.de
[2] RWTH Aachen<first>.<last>@rwth-aachen.de
[3] Fraunhofer Institute for Applied Information Technology

**Abstract.** The models of the interconnected European power system is
exchanged in an RDF based format. To ensure stability, a performant
and repeatable evaluation of the data quality is crucial. We will show
how evacuations are possible with semantic web technologies.

## 1   Introduction

The European Network of Transmission System Operators for Electricity (ENTSO-
E) coordinates cross border cooperation, exchange of information and empow-
ers the integration around Europe [3, 4]. The Common Grid Model Exchange
Specification (CGMES) [1] provides the RDF-based exchange format for each
System Operators individual grid model and the combined common grid model.
The ENTOS-Es *Quality of CGMES Datasets and Calculations for System Op-
erations* [2] defines quality requirements onto the grid model using UML and
concerning the RDF schema. These requirements are ordered in levels from one
to seven with different targets. The first three address file structure and naming
conventions, the following two define constraints to objects and the consistency
– this is the focus of this work. The last two address robustness and cross IGM
inconsistencies. They are the object of ongoing development.

   The joint project *Redispatch-Ermittlungs-Server* of SOPTIM AG and FGH
GmbH includes a quality evaluation for CGMES data. The *Redispatch-Ermittlungs-
Server* uses semantic web technologies for data handling, querying and evaluat-
ing manipulations. We will transform UML invariants and RDF schema require-
ments into SHACL [5] shapes. Further, we demonstrate how these shapes can
help maintain the quality of CGMES data.

## 2   Preventing quality degradation despite data set changes and enhancements

Levels four and five of the Quality of CGMES [2] defines three groups of require-
ments onto the model:

1. Multiplicity of properties are bound by the definition, multiplicity means
   triple with same subject and property but different object

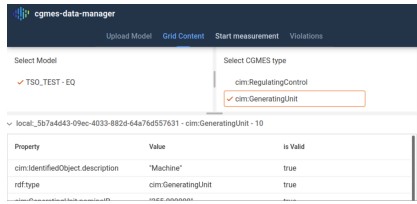

**Fig. 1.** Analysis report after automatic import in data manager interface

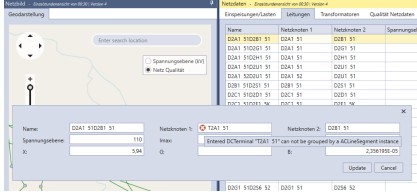

**Fig. 2.** Indication of false grouping in the engineers interface

2. Properties are defined type-safe, holds for literals and references properties
3. Grouping of instances is only possible under certain group types

The first two requirements are included in the RDF schema definition via standard and custom properties. For the multiplicity the ENTSO-E uses the schema property `cims:multiplicity` with values `rdfs:M:1..1` and `rdfs:M:0..1` from RDFS extension. SHACL can check the multiplicities with the cardinality constraint components `sh:maxCount` and `sh:minCount`. To express the type-save requirement the ENTSO-E use `cims:dataType` to indicate the xsd-datatype of a literal and `rdfs:range` for references. In SHACL `sh:class` indicates the type of a reference and `sh:datatype` of a literal.

For the last requirement, the information is present as invariants to the UML classes. Logical and arithmetical expressions are easy to transform into SHACL using value range and logical constraint components. The use of the `<p>.oclIsKindOf(<t>))` method is the only more complex situation, they mean that the end of the path `p` is of type `t`. With SHACL Property Paths in combination with `sh:hasValue` it is possible to mimic this semantic with SHACL.

The SHACL shapes are capable of detecting requirements violations. Fig. 1 shows an analysis result after importing a grid. The isValid column indicates that if the property fulfils the multiplicity and datatype requirements, violations of the grouping requirements are visible in the violations view. The red cross in Fig. 2 warns the engineer that the current value violates the grouping requirement and needs reconsideration.

## 3   Lessons Learned and Further Work

SHACL is not sufficient for the lower three CGMESs quality levels because filenames and structures are out of SHACLs scope. The upper two levels target robustness in terms of robust calculations and interconnections between graphs. For these SHACL, we need to extend our approach to work on datasets, but this needs further collaborations.

The evaluation component of the *Redispatch-Ermittlungs-Server* is based on this approach and currently tested by the german TSOs. These tests show that our approach also works for real-world application and data. For the three types of requirement, a working demo is publicly available on one author GitHub[4].

---

[4] https://github.com/MBoegers/shacl-validate-cim

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
