# OpenReview forum: "Maintain electrical power system model quality with Shape Constraints"
_eswc-conferences.org/ESWC/2021/Conference/Industry_Track — Submitted to ESWC 2021 Industry_

### Official Review · ~Josiane_Xavier_Parreira1 · 2021-04-16
**Use of SHACL for quality checks for electrical power networks**

**Rating:** 5
**Confidence:** 4

**Review:**

The paper describes an approach that uses SHACL for check data quality in electrical power networks. It takes as starting point the Common Grid Model Exchange Specification (CGMES) for data exchange among System Operators.

Some of the data requirements defined in the CGMES are translated into SHACL rules, which are then used for data quality checks.

The solution is being implemented in the scope of a joint project called "Redispatch-Ermittlungs-Server". It would be good to include a short description of the companies involved in the project, why are they interest in the solution and how they plan to exploit it. I understand data quality is a generally a desirable, but it would be interesting to know how important this is for their businesses, i.e. how they can benefit from the solution presented.

A few examples seems to show an demo of the application, but a performance analysis as well as the benefits of the approach are not given. The conclusion suggests that SHACL is not enough to represent all requirements. It is not clear whether this needs to be addressed before the solution can be put into practice.

Also, please check the abstract for grammar and misspellings.

---

### Official Review · ~Aparna_Saisree_Thuluva2 · 2021-04-19
**Maintain electrical power system model quality with Shape Constraints**

**Rating:** 6
**Confidence:** 4

**Review:**

The paper is a good example of using semantic web technologies such as ontologies and SHACL shapes in the real-world use cases. However the approach uses very basic and core features of SHACL for basic validation. SHACL also provides advanced features such as SHACL-SPARQL, SCHAL JS functions etc which can be used for complex validations. Therefore, the maturity of implemention can be improved.
The quality of description should be significantly improved, especially the lessons learned are currently described vaguly. If the quality of description is improved then the paper can be considered for acceptance.

---

### Official Review · ~Konrad_Diwold1 · 2021-04-21
**Review: Maintain electrical power system model quality with Shape Constraints**

**Rating:** 5
**Confidence:** 3

**Review:**

In their article, the authors demonstrate how SHACL can be utilized to maintain the quality of ENTSO-E Common Grid Model Exchange Specification (CGMES) data.

My main question regarding this contribution is how it compares to

- "Nenadić, Kosa R., Milan M. Gavrić, and Vladimir I. Đurđević. "Validation of CIM datasets using SHACL." 2017 25th Telecommunication Forum (TELFOR). IEEE, 2017."

or

- "Larhrib, M., Escribano, M., Cerrada, C., & Escribano, J. J. (2020). Converting OCL and CGMES Rules to SHACL in Smart Grids. IEEE Access, 8, 177255-177266."

the approaches seem similar.

The presented learnings from an industry point of view are rather vague "The evaluation component of the Redispatch-Ermittlungs-Server is based on this approach and currently tested by the german TSOs. These tests show that our approach also works for real-world application and data." - it's not clear what that means precisely. Questions a reader might ask her/himself: how often do such errors (requirement violations) occur during operation? what error finding speeding up is achieved? is the approach accepted by the engineers?, are there plans to integrate it in the TSO processes? are there any barriers to such an integration? Also, more details on future work regarding the planned extension towards datasets would be interesting.

---

### Official Review · ~Victor_Charpenay1 · 2021-04-23
**Relevant submission; writing could be improved**

**Rating:** 7
**Confidence:** 4

**Review:**

The submission introduce a use case for SHACL in the energy domain. OCL constraints defined by the CIM standard are turned into SHACL, for use in an application to manage energy grids. This kind of work definitely has a place at the Industry track of ESWC. Writing could be improved, though.

Ideas for improvement:
 - the first sentence refers to "the integration" but doesn't say what is being integrated. It also doesn't say in what domain the "information" is being exchanged.
 - "for each System Operators individual grid model": how to parse that expression? Should it be "System Operator's individual grid model"? Later in the paper, we find acroyms IGM and TSO. I believe they refer to individual grid models and transmission system operators. Because the abstract is short and most readers won't know the domain of application, I suggest not to use acronyms and replace them with "grid model" and "system operators". The two notions could be shortly introduced at the beginning.
 - "requirements onto" is used twice in the abstract but that preposition is uncommon. Use "requirements for" instead? The first time that expression is used, it is followed by "concerning the RDF schema". I don't know how to read that.
 - "We will transform UML invariants": why is "will" used here? Hasn't it already been done? The rest of the sentence is in present tense.
 - after reading the paper, I still don't get what "grouping of instances" is (third group of CGMES requirements). An example in the extended abstract would help.

Minor comments:
 - "Redispatch-Ermittlungs-Server": provide an English translation
 - "type-save": "type-safe"